



# Measurement of interferences associated with the detection of the hydroperoxy radical in the atmosphere using laser-induced fluorescence

Michelle M. Lew[1], Sebastien Dusanter[2,3], and Philip S. Stevens[1,3]

(1)  Department of Chemistry, Indiana University, Bloomington, IN, USA
(2)  IMT Lille Douai, Univ. Lille, SAGE - Département Sciences de l'Atmosphère et Génie de l'Environnement, 59000 Lille, France
(3)  School of Public and Environmental Affairs, Indiana University, Bloomington, IN, USA

*Correspondence to*: Philip S. Stevens (pstevens@indiana.edu)

**Abstract.** One technique used to measure concentrations of the hydroperoxy radical ($HO_2$) in the atmosphere
involves chemically converting it to OH by addition of NO and subsequent detection of OH.  However, some
organic peroxy radicals ($RO_2$) can also be rapidly converted to $HO_2$ (and subsequently OH) in the presence of
NO, interfering with measurements of ambient $HO_2$ radical concentrations. This interference must be
characterized for each instrument to determine to what extent various $RO_2$ radicals interfere with measurements
of $HO_2$ and to assess the impact of this interference on past measurements. The efficiency of $RO_2$ to $HO_2$
conversion for the Indiana University Laser-Induced Fluorescence – Fluorescence Assay by Gas Expansion (IU-
FAGE) instrument was measured for a variety of $RO_2$ radicals. Known quantities of OH and $HO_2$ radicals were
produced from the photolysis of water vapor at 184.9 nm, and $RO_2$ radicals were produced by the reaction of
several volatile organic compounds with OH. The conversion efficiency of $RO_2$ radicals to $HO_2$ was measured
when NO was added to the sampling cell for conditions employed during several previous field campaigns. For
these conditions, approximately 80% of alkene derived $RO_2$ radicals and 20% of alkane derived $RO_2$ radicals were
converted to $HO_2$. Based on these measurements, interferences from various $RO_2$ radicals contributed to
approximately 35% of the measured $HO_2$ signal during the Mexico City Metropolitan Area (MCMA) 2006
campaign, where the measured VOCs consisted of a mixture of saturated and unsaturated species. However, this
interference can contribute more significantly to the measured $HO_2$ signal in forested environments dominated by
unsaturated biogenic emissions such as isoprene.





**1 Introduction**
The hydroxyl radical (OH) is one of the primary oxidants in the atmosphere (Levy, 1972). The reaction of OH
radicals with volatile organic compounds (VOCs) leads to the production of peroxy radicals, both the hydroperoxy
radical ($HO_2$) and organic peroxy radicals ($RO_2$), which in the presence of nitrogen oxides ($NO_x = NO + NO_2$)
can lead to the production of ozone and secondary organic aerosols in the atmosphere. As a consequence, the
development of effective control strategies for the formation of these pollutants requires an accurate understanding
of the OH, $HO_2$, and $RO_2$ radical chemistry in the atmosphere. Measurements of OH and $HO_2$ (together $HO_x$) can
provide a robust test of our understanding of this complex oxidation chemistry.
Multiple field campaigns have been conducted over the years measuring OH and $HO_2$ radicals in both
urban and forested environments. While much attention has been focused on discrepancies between measured and
modeled OH concentrations (Rohrer et al., 2014), the agreement between measured and modeled $HO_2$
concentrations has been highly variable. In urban environments, measured $HO_2$ concentrations were sometimes
found to agree with model predictions (Shirley et al., 2006; Emmerson et al., 2007; Dusanter et al., 2009b;
Michoud et al., 2012; Lu et al., 2013; Ren et al., 2013; Griffith et al., 2016), while other times the measurements
were found to be both lower (George et al., 1999; Konrad et al., 2003) and higher than model predictions (Martinez
et al., 2003; Ren et al., 2003; Emmerson et al., 2005; Kanaya et al., 2007a; Chen et al., 2010; Sheehy et al., 2010;
Czader et al., 2013; Griffith et al., 2016). In forested environments, measured $HO_2$ concentrations were sometimes
found to agree with model predictions (Tan et al., 2001; Ren et al., 2005; 2006), but were often found to be either
lower (Carslaw et al., 2001; Kanaya et al., 2007b; Whalley et al., 2011; Kanaya et al., 2012; Mao et al., 2012;
Griffith et al., 2013), or higher than model predictions (Carslaw et al., 2001; Kubistin et al., 2010; Kim et al.,
2013; Hens et al., 2014).
These results question our understanding of $HO_x$ radical chemistry and the ability of models to simulate
future changes in the chemical composition of the atmosphere. However, a recent intercomparison of several
instruments measuring $HO_2$ found that the agreement between the different instruments was variable, although
the measurements were highly correlated (Fuchs et al., 2010). While the differences were within the combined
uncertainties of the measurements, there were several measurement periods when the differences could not be
explained by instrumental uncertainties. These results suggested the possibility of potential interferences in the
$HO_2$ measurement technique.
Laser-induced fluorescence using the Fluorescence Assay by Gas Expansion technique (LIF-FAGE) is a
common method for measuring $HO_2$ radicals in the atmosphere. In this technique $HO_2$ radicals are measured





indirectly after sampling ambient air at low pressure through chemical conversion to OH by addition of NO as
shown in reaction R1 and subsequent detection of OH by LIF:

3        $$HO_2 + NO \rightarrow OH + NO_2 \tag{R1}$$

It was previously believed that the detection of $HO_2$ radicals using the LIF-FAGE technique was free from
interferences from the reaction of $RO_2$ radicals with NO, as model simulations and measurements suggested that
the rate of conversion of $RO_2$ radicals to $HO_2$ by reactions R2 and R3 and subsequent conversion to OH through
reaction R1 was negligible due to the slow rate of reaction R3 under the reduced oxygen concentration in the low
pressure LIF-FAGE cell and the short reaction time between injection of NO and detection of OH (Heard and
Pilling, 2003).

10        $$RO_2 + NO \rightarrow RO + NO_2 \tag{R2}$$

11        $$RO + O_2 \rightarrow R'O + HO_2 \tag{R3}$$

For example, $RO_2$ radicals produced from the OH-initiated oxidation of small alkanes were found to produce a
negligible yield of $HO_2$ (Stevens et al., 1994; Kanaya et al., 2001; Tan et al., 2001; Creasey et al., 2002; Holland
et al., 2003).
However, recent laboratory studies have shown that there are interferences associated with measurements
of $HO_2$ using this technique from the conversion of $RO_2$ radicals derived from the OH-initiated oxidation of
alkenes and aromatics to $HO_2$ (and subsequently OH) by reaction with NO. Measured $RO_2$ to $HO_2$ conversion
efficiencies of 95% for the peroxy radicals derived from the OH-initiated oxidation of propene and 86% for the
peroxy radicals derived from the OH-initiated oxidation of benzene have been reported (Fuchs et al., 2011). The
high conversion efficiency of alkene-based peroxy radicals to $HO_2$ is due to the ability of the β-hydroxyalkoxy
radicals produced from reaction R2 to rapidly decompose forming a hydroxyalkyl radical which then reacts rapidly
with $O_2$ leading to the production of a carbonyl compound and $HO_2$ (Fuchs et al., 2011; Whalley et al., 2013). The
conversion efficiency depends on the instrumental characteristics and the configurations employed (Fuchs et al.,
2011; Whalley et al., 2013). As a result, this interference must be characterized for all LIF-FAGE instruments for
the accurate analysis of ambient $HO_2$.
This paper will describe the characterization of the $RO_2$ interferences associated with the Indiana
University LIF-FAGE instrument under several past campaign configurations. These include the Mexico City
Metropolitan Area (MCMA) campaign in 2006 (Dusanter et al., 2009a; 2009b), the Community Atmosphere-
Biosphere INteractions EXperiment (CABINEX) in 2009 (Griffith et al., 2013), and the California Research at
the Nexus of Air Quality and Climate Change campaign in Los Angeles (CalNex-LA) in 2010 (Griffith et al.,



2016). The impact of this interference on the previously published results from the MCMA-2006 campaign and a
reanalysis of these $HO_2$ measurements will be discussed.
**2 Experimental Section**
**2.1 Instrument description**
The Indiana University LIF-FAGE instrument (IU-FAGE) has been described in detail previously (Dusanter et
al., 2008; 2009a; Griffith et al., 2013; 2016). In the LIF-FAGE technique, OH radicals are detected by laser-
induced fluorescence after expansion of ambient air to low pressure. This enhances the OH fluorescence lifetime,
allowing temporal filtering of the fluorescence from laser scatter (Heard and Pilling, 2003). A diagram of the IU-
FAGE detection cell is illustrated in Fig. 1. Ambient air is expanded through an orifice between 0.635 mm and
1.016 mm diameter located at the top of a cylindrical nozzle (5 cm in diameter and 20 cm long). The size of the
orifice was kept unchanged during each campaign but was varied between the different campaigns reported here.
Two scroll pumps (Edwards XDS 35i) connected in parallel maintain a pressure inside the cell between 4 and 7.5
Torr depending on the sampling size of the orifice and the pumping speed, resulting in a flow rate between 3 and
10 SLPM through the sampling nozzle.

15       The original IU-FAGE laser system used in this study consisted of a Spectra Physics Navigator II

YHP40-532Q diode-pumped Nd:YAG laser that produced approximately 5.5W of radiation at 532 nm at a
repetition rate of 5 kHz. This laser pumped a Lambda Physik Scanmate 1 dye laser (Rhodamine 640, 0.25 g L$^{-1}$
in isopropanol) that produced tunable radiation around 616 nm, which was frequency doubled to generate 2 to 20
mW of radiation at 308 nm (~20 ns pulse width). This laser system was recently replaced with a Spectra Physics
Navigator II YHP40-532Q that produces approximately 8 W of radiation at 532 nm at a repetition rate of 10 kHz
that pumps a Sirah Credo Dye laser (255 mg/L of Rhodamine 610 and 80 mg/L of Rhodamine 101 in ethanol),
resulting in 40 to 100 mW of radiation at 308 nm. After exiting the dye laser, the beam was focused onto the
entrance of a 12 m optical fiber to transmit the radiation to the sampling cell. In the detection cell, the laser crosses
the expanded air perpendicular to the flow in a White cell configuration with approximately 24 passes.

25       OH radicals are excited using the $A^2\Sigma^+\ \upsilon'=0 \leftarrow X^2\Pi\ \upsilon''= 0$ transition near 308 nm (Stevens et al., 1994).

The net signal is measured by turning the wavelength on- and off-resonance in successive modulation cycles. A
reference cell where OH is produced by thermal dissociation of water vapor is used to ensure that the laser is tuned
on and off the OH transition. The OH fluorescence is detected using a microchannel plate photomultiplier tube
(MCP-PMT) detector (Hamamatsu R5946U-50), a preamplifier (Stanford Research System SR445) and a gated

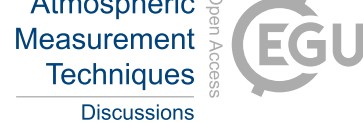

photon counter (Stanford Research Systems SR 400). The MCP-PMT is switched off during the laser pulse
through the use of electronic gating allowing the OH fluorescence to be temporally filtered from laser scattered
light.
A Teflon injector located approximately 2.5 cm below the inlet and 17.5 cm above the detection axis
(Fig. 1) allowed for the addition of NO (Matheson, 99.8%) to convert ambient $HO_2$ to OH through reaction R1.
The fraction of $HO_2$ ($C_{HO2}$) converted into OH was measured during calibration experiments (Dusanter et al.,
2008). The NO flow (approximately $1$-$3 \times 10^{13}$ cm$^{-3}$) maximized the conversion of $HO_2$ into OH while minimizing
the removal of OH by the OH + NO reaction.
**2.2 Instrument Calibration for OH and HO₂**
The IU-FAGE instrument is calibrated by producing known quantities of OH and $HO_2$ radicals from the photolysis
of water vapor in air (reactions R4 and R5) (Dusanter et al., 2008):
$$H_2O + hv \text{ (184.9 nm)} \rightarrow H + OH \tag{R4}$$
$$H + O_2 \rightarrow HO_2 \tag{R5}$$
The calibration source consists of an aluminum flow reactor ($1.27 \times 1.27 \times 30$ cm) equipped with quartz windows
on two sides (Fig. 2). The light source consists of a low-pressure mercury lamp (UVP Inc.) housed in an aluminum
cartridge that is continuously purged with dry nitrogen to prevent light absorption by gases in addition to helping
to stabilize the lamp temperature. The radiation from the lamp passes through a bandpass filter centered at 185
nm (Acton Research) prior to entering the reactor and is detected by a photodiode. The lamp housing can be
adjusted along the length of the calibrator to measure the loss of radicals between the source region and the exit
of the calibrator.
The concentration of OH and $HO_2$ radicals produced by the calibration source can be determined from
the following equation:
$$[OH] = [HO_2] = [H_2O] \cdot \sigma_{H_2O} \cdot \varphi_{OH+H} \cdot F \cdot t \tag{1}$$
In this equation $\varphi_{OH+H}$ is the quantum yield of OH from water photolysis, and $\sigma_{H2O}$ is the absorption cross section
of water ($7.14 \times 10^{-20}$ cm$^{-2}$ molecule$^{-1}$ (Cantrell et al., 1997; Hofzumahaus et al., 1997; Creasey et al., 2000)). The
product of the photon flux (F) and the photolysis time (t) can be determined from oxygen actinometry, as the
photolysis of oxygen at 185 nm leads to the production of ozone (reactions R6 and R7) (Okabe, 1978):
$$O_2 + hv \text{ (184.9 nm)} \rightarrow 2O(^3P) \tag{R6}$$





$$O_2 + O(^3P) + M \rightarrow O_3 + M \qquad \text{(R7)}$$

The concentration of $HO_x$ radicals can thus be calculated from measured concentrations of water and ozone using
Eq. (2) (Heard and Pilling, 2003; Holland et al., 2003):

$$[OH] = [HO_2] = [H_2O] \cdot \sigma_{H_2O} \cdot \varphi_{OH+H} \cdot \frac{[O_3]}{\varphi_{O_3} \cdot \sigma_{O_2} \cdot [O_2]} \qquad \text{(2)}$$

Here $\varphi_{O3}$ is the quantum yield of $O_3$ from oxygen photolysis and and $\sigma_{O2}$ is the absorption cross sections of $O_2$,
which must be experimentally determined for each penlamp due to the overlap of the highly structured absorption
spectrum of $O_2$ and the lineshape of the emission at 184.9 nm. The lineshape depends on the operating conditions
of the lamp as a result of line reversal and potential fluorescence of the fused silica envelope (Cantrell et al., 1997;
Hofzumahaus et al., 1997; Lanzendorf et al., 1997).
**2.3 Measurement of the RO₂ conversion efficiency to HO₂**
Various alkenes (isoprene, methyl vinyl ketone, methacrolein, methyl ethyl ketone, ethene, trans-2-butene,
tetramethylethylene), alkanes (propane, butane, octane), and aromatic compounds (toluene) were used to measure
the conversion efficiency of $RO_2$ radicals to $HO_2$. These VOCs were added to the main calibrator flow
approximately 190 ms prior to the radical source to ensure that the added VOC was well mixed into the humid air
flow before photolysis within the calibration source. The concentration of each VOC added to the calibrator was
increased to react and remove the majority of the OH produced in the calibrator, resulting in $RO_2$ concentrations
that were approximately equal to the concentration of OH reacted away. These $RO_2$ radicals are then sampled into
the IU-FAGE instrument. Addition of NO inside the detection axis converts a fraction of the $RO_2$ radicals to $HO_2$
through reactions R2 and R3. Since $RO_2$ is produced together with $HO_2$ in the calibrator, there is a subsequent
conversion of both $RO_2$ and $HO_2$ into OH in the IU-FAGE cell, which is then detected by LIF.
Figure 3 illustrates two typical experiments designed to measure the conversion efficiency of $RO_2$
radicals to $HO_2$ in the IU-FAGE instrument. The total $HO_x$ signal is defined as the sum of the total OH ($S_{OH}$) and
$HO_2$ ($S_{HO2}$) produced by the mercury penlamp in the absence of the added VOC (Eq. (3)):

$$S_{HO_x} = S_{HO_2} + S_{OH} \qquad \text{(3)}$$

The OH concentration produced by the penlamp is measured at the beginning, middle, and at the end of each
experiment to ensure that the concentrations remained stable (experimental mode 1 in Fig. 3). Once the OH signal



($S_{OH}$) stabilizes, NO is added internally to the detection cell to convert $HO_2$ into OH and measure the total $HO_x$
signal ($S_{HOx}$) (mode 2 in Fig. 3). The conversion efficiency of $HO_2$ to OH is defined by Eq. (4):

$$C_{HO_2 \rightarrow OH} = \frac{S_{HO_x} - S_{OH}}{S_{OH_0}} = \frac{S_{HO_2}}{S_{OH_0}} \tag{4}$$

$S_{OH0}$ is the OH signal after accounting for the loss of OH on the walls of the calibrator (approximately 20%). The
wall loss for $HO_2$ is negligible in the calibrator (Dusanter et al., 2008).

6        Next, internal NO addition is stopped and the OH signal is measured again to ensure the stability of

radical production during the experiment. The VOC is then added to the calibration system resulting in a decrease
in the observed OH signal (mode 3 in Fig. 3). The remaining OH signal in the presence of the VOC is denoted as
$S_{OH+VOC}$. For alkenes such as isoprene, the fast reaction with OH results in an almost total removal of OH radicals
from the calibration source and $S_{OH+VOC}$ is close to zero. However, for less reactive alkanes such as butane, the
added VOC concentration was often not sufficient to completely remove OH radicals due to the short reaction
time in the calibrator, resulting in a non-zero $S_{OH+VOC}$ signal. The conversion efficiency in which OH radicals are
converted to $RO_2$ radicals ($C_{OH+VOC}$) is defined by Eq. (5), derived from integrating the expressions for the rate of
OH loss and the rate of $RO_2$ production from the OH +VOC reaction:

$$C_{OH+VOC} = \frac{[RO_2]}{[OH]_0} = \frac{F_{OH} - k_w t}{F_{OH}}(1 - e^{-F_{OH}}) \qquad F_{OH} = ln\left(\frac{S_{OH_0}}{S_{OH+VOC}}\right) \tag{5}$$

Here $k_w t$ is the product of the rate constant for reaction of OH radicals on the wall of the calibration source with
the reaction time $t$, reflecting the measured loss of OH on the walls of the calibrator (Dusanter et al., 2008).

18        The subsequent addition of NO to the detection cell will convert a fraction of $RO_2$ radicals and $HO_2$

radicals to OH (mode 4 in Fig. 3). The conversion efficiency of $RO_2$ to OH ($C_{RO2 \rightarrow OH}$) is determined by
multiplying the fraction of $RO_2$ radicals converted to $HO_2$ ($f_{RO2 \rightarrow HO2}$) with the conversion efficiency of $HO_2$ to
OH ($C_{HO2 \rightarrow OH}$):

$$C_{RO_2 \rightarrow OH} = f_{RO_2 \rightarrow HO_2} \cdot C_{HO_2 \rightarrow OH} \tag{6}$$

The signal due to $RO_2$ radicals ($S_{RO2}$) is defined as the original OH signal ($S_{OHo}$) multiplied by the conversion
efficiency of OH radicals to $RO_2$ radicals ($C_{OH+VOC}$) and multiplied by the conversion efficiency of $RO_2$ to OH
($C_{RO2 \rightarrow OH}$) (Eq. (7)):

$$S_{RO_2} = S_{OH_0} \cdot C_{OH+VOC} \cdot C_{RO_2 \rightarrow OH} \tag{7}$$



For OH +VOC reactions that lead to the production of $HO_2$ with a yield of $y$ (OH + benzene and toluene for
example (Klotz et al., 1998; Nehr et al., 2011)), the OH to $RO_2$ conversion efficiency ($C_{OH+VOC}$) must be multiplied
by the overall yield ($1-y$) of $RO_2$ radicals produced from the OH +VOC reaction. Taking this yield into account,
the signals due to $RO_2$ and $HO_2$ radicals become:

$$S_{RO_2} = S_{OH_0} \cdot C_{OH+VOC} \cdot (1-y) \cdot C_{RO_2 \rightarrow OH} \tag{7a}$$

$$S_{HO_2\ total} = S_{OH_0} \cdot C_{OH+VOC} \cdot y \cdot C_{HO_2 \rightarrow OH} + S_{HO_2} \tag{8}$$

The measured OH signal under these conditions ($S_{ROx}$) reflects the contribution of $RO_2$, $HO_2$, and unreacted OH
radicals (experimental mode 4):

$$S_{RO_x} = S_{RO_2} + S_{HO_2\ total} + S_{OH+VOC} \tag{9}$$

$$S_{RO_x} = \left( S_{OH_0} \cdot C_{OH+VOC} \cdot (1-y) \cdot C_{RO_2 \rightarrow OH} \right) + \left( S_{OH_0} \cdot C_{OH+VOC} \cdot y \cdot C_{HO_2 \rightarrow OH} + S_{HO_2} \right) + S_{OH+VOC} \tag{9a}$$

Combining equations 3, 6, and 9a results in an expression for the fraction of $RO_2$ radicals converted to $HO_2$
($f_{RO2 \rightarrow HO2}$) that can be expressed as the measured signals for each experimental mode ($S_{OH}$, $S_{HOx}$, $S_{OH+VOC}$,
$S_{ROx}$) as seen in Eq. (10):

$$f_{RO_2 \rightarrow HO_2} = \frac{S_{RO_x} - S_{HO_x} + S_{OH} - S_{OH+VOC} - S_{OH_0} \cdot C_{OH+VOC} \cdot y \cdot C_{HO_2 \rightarrow OH}}{S_{OH_0} \cdot C_{OH+VOC} \cdot (1-y) \cdot C_{HO_2 \rightarrow OH}} \tag{10}$$

When the yield of $HO_2$ from the OH + VOC reaction is zero ($y = 0$), and under conditions where all the OH
radicals are converted to $RO_2$ ($S_{OH+VOC} = 0$), the above equation (with Eq. 3 and 4) simplifies to the following:

$$f_{RO_2 \rightarrow HO_2} = \frac{S_{RO_x} - S_{HO_x} + S_{OH}}{S_{OH_0} \cdot C_{OH+VOC} \cdot C_{HO_2 \rightarrow OH}} = \frac{S_{RO_x} - S_{HO_x} + S_{OH}}{(S_{HO_x} - S_{OH}) \cdot C_{OH+VOC}} \tag{11}$$

Because this method cannot distinguish between the different peroxy radicals that could be produced from each
OH + VOC reaction, the measured conversion efficiency reflects the average conversion efficiency of all peroxy
radicals for a given VOC.
**3 Results**
The pressure and flow conditions for the three campaigns conducted with the IU LIF-FAGE instrument are
summarized in Table 1. For each characterization, the flow rate of NO addition was kept constant at 1 sccm in



order to determine the impact of the different operating conditions on the measured $RO_2$-to-$HO_2$ conversion
efficiencies. This is the NO flow rate used during the MCMA-2006, CABINEX and CalNex campaigns, and
resulted in $HO_2$-to-OH conversion efficiencies that were similar to that measured during both the CABINEX and
the CalNex campaigns. However, the measured $HO_2$-to-OH conversion efficiency for the MCMA-2006 campaign
configuration in these experiments was approximately 20% lower than that previously reported (Dusanter et al.,
2008; 2009a). The reason for this discrepancy is unclear, and may indicate problems in precisely recreating the
flow conditions during this campaign in these laboratory experiments. In addition, the NO flow rate was varied
during MCMA-2006 in order to maximize the $HO_2$-to-OH conversion efficiency and to quantify the photolytic
interference associated with high NO concentrations in the detection cell. Thus is possible that the actual flow rate
used to maximize the conversion efficiency was slightly greater than the 1 sccm reported. As a result, the
conversion efficiencies measured in this study for the MCMA-2006 configuration may represent a lower limit to
the actual conversion efficiencies during the campaign.

13        The $RO_2$ conversion efficiency into $HO_2$ ($f_{RO2 \rightarrow HO2}$) measured for the inlet conditions for the MCMA

2006, CABINEX, and CalNex campaigns are summarized in Table 2 and represent the results of several
experiments similar to those illustrated in Fig. 2, with the uncertainty representing one standard error of the mean
of the measurements. The largest $RO_2$ interference was observed for the CalNex inlet conditions where alkenes
produced interferences ranging from $83 \pm 7\%$ for isoprene-based peroxy radicals to $96 \pm 6\%$ for
tetramethylethylene (TME)-based peroxy radicals, while the conversion efficiency of aromatic, aldehydes, and
ketone compounds ranged from $54 \pm 4\%$ for methacrolein (MACR) to $91 \pm 4\%$ for methyl vinyl ketone (MVK).
The $RO_2$ to $HO_2$ conversion efficiency of a number of alkanes ranged from an average measured value of $15 \pm$
$3\%$ for propane-based peroxy radicals to $62 \pm 4\%$ for octane-based peroxy radicals, with the $RO_2$ to $HO_2$
conversion efficiency increasing with the carbon number. The inlet configuration and conditions used during the
MCMA 2006 campaign generally resulted in lower $RO_2$ interferences likely due to the higher flow rate (and
shorter reaction time) in the detection cell and the lower NO concentration, although the measured conversion
efficiency was found to be somewhat greater for some VOCs. Under these inlet conditions the $RO_2$ to $HO_2$
conversion efficiency for propane-based peroxy radicals was measured to be $22 \pm 11\%$ while the conversion
efficiency for octane-based peroxy radicals was $30 \pm 5\%$. Because the CABINEX campaign occurred in a remote
forested environment, measurements of the $RO_2$-to-$HO_2$ conversion efficiency focused on characterizing
interferences from peroxy radicals produced from isoprene and its oxidation products (MVK and MACR), as
isoprene peroxy radicals were predicted to contribute to more than 80% of the total $RO_2$ concentration during the
campaign (Griffith et al., 2013). The inlet and instrumental configuration during CABINEX resulted in a higher





pressure and slower sampling rate compared to the MCMA 2006 configuration. For this instrumental
configuration, the $RO_2$-to-$HO_2$ conversion efficiency was found to be $91 \pm 5\%$ for isoprene-based peroxy radicals,
while the conversion efficiencies for MVK and MACR were found to be $62 \pm 5\%$ and $30 \pm 7\%$, respectively.
These observations are consistent with results reported for other FAGE instruments (Fuchs et al., 2011;
Whalley et al., 2011), and assumes that the photolysis of each VOC does not contribute to the production of
radicals in these experiment. However, tests to determine whether photolysis of the various VOCs resulted in the
formation of $HO_x$ radicals in the absence of water vapor revealed that the photolysis of methyl vinyl ketone
(MVK), methacrolein (MACR), methyl ethyl ketone (MEK), and toluene can lead to the production of $HO_x$
radicals. The radical signals from the photolysis of methacrolein, and toluene were small and negligible relative
to the total $HO_x$ signal produced from the photolysis of water. However, the signal from the photolysis of MVK
and MEK during these tests was significant and could interfere with the measurements of the $RO_2$-to-$HO_2$
conversion efficiency. These results are in contrast to that reported by Fuchs et al. (2011), who found that the
photolysis of VOCs during similar tests in dry air did not produce any radicals. The interference from the radicals
produced from the photolysis of MVK and MEK would result in higher apparent conversion efficiencies, and
could contribute to the higher $RO_2$-to-$HO_2$ conversion efficiency reported here for MVK compared to that reported
by Fuchs et al. (2011).
As previously observed, the $RO_2$-to-$HO_2$ conversion efficiency of alkene-based β-hydroxyalkyl peroxy
radicals was found to be greater than the conversion efficiency of alkane-based alkyl peroxy radicals (Fuchs et
al., 2011). As discussed above, this is due to due to the ability of the β-hydroxyalkoxy radicals produced from the
$RO_2$ + NO reaction to rapidly decompose to form a hydroxyalkyl radical that can reacts rapidly with $O_2$ in the
FAGE detection cell leading to the production of a carbonyl compound and $HO_2$. However, the ability of large
alkoxy radicals to rapidly isomerize and decompose also results in a rapid production of $HO_2$ radicals and a larger
conversion efficiency.
In general, reducing the reaction time in the IU-FAGE instrument reduces the conversion of these peroxy
radicals to $HO_2$, as illustrated by the reduced conversion efficiencies between the CalNex and MCMA operating
conditions for the majority of the VOCs tested. However, the measured conversion efficiencies of some of the
tested VOCs did not always display this behavior and the reasons for the discrepancies are unclear. For example,
the conversion efficiency for ethene peroxy radicals was lower for the CalNex configuration compared to the
CABINEX and MCMA configurations even though the overall flow rate was slower for the CalNex configuration.
However, the $HO_2$-to-OH conversion efficiency was also lower for this inlet configuration, suggesting that
reaction time may not be the only factor limiting the conversion efficiency under these instrument conditions.




Similarly, the conversion efficiency of MVK and MACR measured for the CABINEX instrument configuration
was lower than that measured for the MCMA inlet configuration, even though the overall slower flow rate in the
CABINEX configuration leads to a longer reaction time in the IU-FAGE detection cell. This may suggest that the
chemistry of peroxy radicals produced from the OH-initiated oxidation of MVK and MACR is different than that
of the peroxy radicals produced from the OH-initiated oxidation of alkenes and alkanes (Fuchs et al., 2011).
**4 Discussion**
**4.1 $RO_2$ Radical Concentrations during MCMA 2006**
The previous analysis of the $HO_2$ radical concentrations during the Mexico City Metropolitan Area (MCMA)
2006 did not take into account interferences from $RO_2$ radicals (Dusanter et al., 2009b). As discussed above, the
instrumental conditions during MCMA-2006 resulted in the conversion of a fraction of $RO_2$ radicals to $HO_2$,
resulting in the measurements reflecting $HO_2^* = HO_2 + \alpha RO_2$ and overestimating the actual $HO_2$ concentrations.
To determine the fraction ($\alpha$) of $RO_2$ radicals likely detected during the $HO_2$ measurements, the $RO_2$ radical
concentrations during MCMA-2006 that were previously modeled using the Regional Atmospheric Chemistry
Mechanism (RACM) were used to calculate the modeled $HO_2^*$ concentrations (Dusanter et al., 2009b).
As discussed in Dusanter et al. (2009b), the RACM model is a condensed chemical mechanism that
describes the gas-phase oxidation of 17 inorganic and 32 organic species. Kinetic parameters for the reactions of
OH, $O_3$ and $NO_3$ with inorganic species and for reactions involving organic species treated explicitly in RACM
(methane, ethane, ethene, formaldehyde, glyoxal, methyl peroxide and isoprene) were updated using the JPL
database (Sanders et al., 2006). Rate constants and branching ratios for OH, $O_3$ and $NO_3$ reactions with surrogate
species were used as described in the RACM model (Stockwell et al., 1997). Heterogeneous chemistry, such as
the incorporation of trace gases into aerosols, was not included.
The peroxy radical fractions calculated by the model are illustrated in Fig. 4 for 9 am, 12 pm, 6pm (local
times) and the overall diurnal average. Alkane-based peroxy radicals (red shades) include methyl peroxy (RACM
category CH3O2), ethyl peroxy (ETHP), peroxy radicals formed from the oxidation of alkanes, esters, and alkynes
exhibiting OH rate constants lower than $3.4 \times 10^{-12}$ $cm^3$ $molecule^{-1}$ $s^{-1}$ (HC3P), peroxy radicals formed from
alkanes, esters, and alkynes characterized by OH rate constants ranging from $3.4 \times 10^{-12}$ to $6.8 \times 10^{-12}$ $cm^3$
$molecule^{-1}$ $s^{-1}$ (HC5P), and peroxy radicals formed from alkanes, esters, and alkynes whose OH rate constants are
larger than $6.8 \times 10^{-12}$ $cm^3$ $molecule^{-1}$ $s^{-1}$ (HC8P). Alkene-based peroxy radicals (blue shades) include peroxy
radicals from the oxidation of ethene (ETEP), external olefins (OLTP), internal olefins (OLIP), isoprene (ISOP),



and from α-pinene and other cyclic terpenes with one double bond (APIP). Aromatic peroxy radicals (green
shades) include species produced during the oxidation of toluene (TOLP), xylenes (XYLP), and cresol (CSLP).
The carbonyl-based peroxy radicals (grey shades) include saturated (ACO3) and unsaturated (TCO3) acyl peroxyl
radicals.

5         The total average modeled $RO_2$ concentration from 9:00 am to 6:00 pm consisted of 54% alkane-based,

27% alkene-based, and 14% aromatic-based peroxy radicals (Fig. 4). On average, the modeled composition of
peroxy radicals was relatively constant throughout the day during the MCMA campaign. The modeled relative
contribution of aromatic-based peroxy radicals was greater in the morning, consistent with the observed elevated
concentrations of benzene and toluene during the morning hours (Dusanter et al., 2009b).
**4.2 Implications of $RO_2$ interferences for $HO_2$ measurements during MCMA 2006**
The modeled diurnal average concentrations of total $RO_2$ radicals during MCMA is shown in Fig. 5, along with
the modeled $HO_2$ concentrations and the measured $HO_2^*$ concentrations. As discussed in Dusanter et al. (2009b),
the modeled $HO_2$ concentrations were in good agreement with the measurements during the afternoon but the
model underestimated the measured $HO_2$ concentrations during the morning hours by a factor of approximately 2
to 5. However, these conclusions were based on the assumption that the measured $HO_2$ concentrations were free
from interferences and could be compared to the modeled $HO_2$ concentrations. Based on the conversion
efficiencies reported for $RO_2$ radicals in the present study, it is clear that the MCMA measurements represent an
upper limit to the actual $HO_2$ concentrations and should be compared to the modeled $HO_2^* = HO_2 + \alpha RO_2$
concentrations.

20        The RACM modeled $HO_2^*$ concentrations were calculated by applying the measured $RO_2$-to-$HO_2$

conversion efficiencies for the instrumental conditions reported in Table 2 for MCMA-2006 using Eq. 12:
$HO_2^* = HO_2 + (0.84 \cdot ISOP + 0.68 \cdot OLIP + 0.68 \cdot OLTP + 0.86 \cdot ETEP + 0.32 \cdot TOLP + 0.32 \cdot XYLP +$
$0.32 \cdot CSLP + 0.72 \cdot APIP + 0.22 \cdot HC3P + 0.22 \cdot HC5P + 0.30 \cdot HC8P + 0.05 \cdot CH3O2 + 0.07 \cdot ETHP$
$+ 0.32 \cdot ACO3 + 0.32 \cdot TCO3 + 0.72 \cdot KETP$                    (12)
The contribution for isoprene peroxy radicals (ISOP), ethene peroxy radicals (ETEP), and toluene peroxy radicals
(TOLP) were taken directly from Table 2. The average $RO_2$-to-$HO_2$ conversion efficiency for *trans*-2-butene and
tetramethylethelene-based peroxy radicals was used for the conversion efficiency of peroxy radicals from internal
olefins (OLIP), and external olefins (OLTP), while the conversion efficiency for *trans*-2-butene was used for the
conversion efficiency for α-pinene and other cyclic terpene peroxy radicals (APIP). The measured conversion



efficiency for toluene-based peroxy radicals was used to represent the conversion efficiency for xylene (XYLP)
and cresol (CSLP) peroxy radicals. The conversion efficiency of methacrolein-based peroxy radicals was used to
represent the conversion efficiency of acetyl peroxy and higher saturated acyl peroxy radicals (ACO3) as well as
unsaturated acyl peroxy radicals (TCO3), while the conversion efficiency of methyl vinyl ketone-based peroxy
radicals was used to represent the efficiency of ketone-based peroxy radicals (KETP).
The overall average contribution of peroxy radicals to the modeled $HO_2$* and the relative contribution of
each RACM peroxy radical category to the $RO_2$ interference are shown in Fig. 6. Because the NO flow rate used
in characterizing the conversion efficiencies in Table 2 was generally lower than the flow rates used during the
campaign, the relative peroxy radical contributions illustrated in this figure are likely lower limits to the actual
contribution during the campaign, as the $HO_2$-to-OH conversion efficiency of 80% in these experiments was
approximately 20% lower than the conversion efficiency measured during the campaign (Dusanter et al., 2008;
Dusanter et al., 2009a).
On average, $RO_2$ radicals contributed to approximately 35% of the total modeled $HO_2^*$ (Fig. 6). While
alkanes compose the majority of the modeled peroxy radicals, they only contributed to about 29% of the $RO_2$
interference, while alkenes contributed to approximately 51% to the interference. The contribution of $RO_2$ radicals
to the measured $HO_2$* concentrations in this environment is in contrast to measurements in forested environments
where the OH reactivity is dominated by isoprene and other unsaturated biogenic emissions. In these
environments, isoprene and other biogenic hydroxyl alkyl peroxy radicals can be the dominant peroxy radicals
and can make a significant contribution to the measured $HO_2$* concentrations due to their high conversion
efficiency to $HO_2$ in the FAGE detection cell (Table 2). For example, during the CABINEX campaign in a northern
Michigan forest, isoprene peroxy radicals were modeled to be the dominant peroxy radical in this environment,
contributing to approximately 50% of the modeled $HO_2$* concentrations during the daytime (Griffith et al., 2013).
As a result, previous measurements of $HO_2$ in these environments by LIF-FAGE or other chemical conversion
techniques are likely influenced by an interference from β-hydroxyalkyl peroxy radicals such as those produced
by the OH-initiated oxidation of isoprene and other biogenic emissions.
The diurnal average modeled $HO_2$* concentrations for the MCMA-2006 campaign are also shown in Fig.
5. As can be seen in this figure, the model overestimates the measured $HO_2$* by approximately 35% between
12:00 and 17:00 CST, although the modeled results are generally close to the upper bound of the calibration
accuracy (36%, 2σ) (Dusanter et al., 2009b). As discussed above, the modeled $HO_2$* is likely a lower limit given
that the $RO_2$-to-$HO_2$ conversion efficiencies during the campaign may be greater than shown in Table 2 due to the
higher NO flows used during the campaign. Although, the measured $HO_2$* are still likely to be within the overall


uncertainty of the model, which was estimated to be approximately a factor of 1.7 (Dusanter et al., 2009b), these
results suggest that the model likely overestimates the measured concentrations during the afternoon.

3       These results are in contrast to the results from the CalNex campaign, where the simulations using the

RACM2 model tended to underestimate the measured $HO_2$* concentrations during the week, when NO mixing
ratios were greater than 4 ppb (Griffith et al., 2016). The reason for this difference between the campaigns is
unclear, but may be related to the relative concentrations of dicarbonyl species and their treatment in the RACM
and RACM2 models. Dusanter et al. (2009b) demonstrated that the RACM model results for MCMA-2006 were
highly sensitive to the concentrations of dicarbonyl species in the model, with the model significantly
overpredicting the concentration of $HO_x$ radicals when unmeasured concentrations of these species were not
constrained. Daytime average measured glyoxal mixing ratios during MCMA-2006 were approximately 0.4 ppb
(Dusanter et al., 2009b), which were greater than the maximum daytime mixing ratios of 0.16 ppb during CalNex
(Washenfelder et al., 2011), suggesting that the MCMA-2006 results may be more sensitive to the treatment of
dicarbonyl chemistry compared to CalNex. Additional analysis and modeling will be needed to resolve this issue.

14       While the model tends to overestimate the measured $HO_2$* concentrations during the afternoon, it

underestimates the measured $HO_2$* concentrations in the morning by a factor of 3 between 9-11 am. As discussed
in Dusanter et al. (2009b), this may suggest that a significant radical source may be missing from current
atmospheric models under polluted conditions. Dusanter et al. (2009b) also compared the measured $HO_2$*/OH
ratio to the RACM modeled $HO_2$/OH ratio and found that the model underpredicted the observed ratio, especially
under conditions where the mixing ratio of NO was greater than 5 ppb. At NO mixing ratios of 10 ppb, the model
underestimated the measured ratio by a factor of 2 (Dusanter et al., 2009b). However, comparing the measured
$HO_2$*/OH ratio to the modeled $HO_2$*/OH ratio improves the agreement even though the model tends to overpredict
both OH and $HO_2$* in the afternoon (Fig. 7). This may indicate that there is either a missing sink of $HO_x$ radicals
in the model or a miscalculation of the relative rates of initiation and/or termination. At an NO mixing ratio of 10
ppb the modeled $HO_2$*/OH ratio is in good agreement with the measurements, although it still underestimates the
measured $HO_2$*/OH ratio at higher NO mixing ratios by as much as a factor of 4, and may also overestimate the
$HO_2$*/OH ratio for mixing ratios of NO less than 5 ppb by as much as a factor of 2 (Fig. 7). It is interesting to note
that a model underestimation of the total OH reactivity at high NO mixing ratios may contribute to this
discrepancy. Unfortunately, total OH reactivity was not measured during MCMA-2006 and the reliability of the
model to simulate it could not be assessed. Similar results were observed for the CalNex campaign (Griffith et al.,
2016), which included direct measurements of the total OH reactivity. Although accounting for the missing
reactivity in the analysis of the CalNex data improved the agreement between the measured and modeled



HO$_2$*/OH ratio, the model still underestimates the measured ratio at high mixing ratios of NO (Griffith et al.,
2016). These results suggest that our understanding of the radical propagation chemistry under high NO conditions
may be incomplete.
**5. Summary and Conclusions**

5        The RO$_2$ interference associated with measurements of HO$_2$ by the IU-FAGE instrument was

characterized for three different instrument configurations that were used in previous field campaigns (MCMA
2006, CABINEX 2009, and CalNex 2010). Similar to that reported for other LIF-FAGE instruments, the RO$_2$-to-
HO$_2$ conversion efficiency was highest for alkene- and aromatic-based RO$_2$ radicals, producing higher levels of
interference, while the conversion efficiency of alkane-based RO$_2$ radicals was less but increased with increasing
carbon number. In general, the conversion efficiency was higher for instrument configurations that involved
slower sampling flow rates and longer reaction times between the peroxy radicals and NO in the detection cell.

12       The similarities in the measured RO$_2$ conversion efficiencies reported here with those reported for other

LIF-FAGE instruments suggest that the main factor controlling the conversion efficiency is the rate of reaction of
RO$_2$ radicals with NO, and that increasing the efficiency of the conversion of HO$_2$ to OH will also increases the
RO$_2$-to-HO$_2$ conversion efficiency. Although the impact of differences in the characteristics of the low pressure
expansion in LIF-FAGE instruments cannot be ruled out, these results suggest that the interferences reported here
associated with measurements of HO$_2$ are likely similar for all instruments that measure HO$_2$ by chemical
conversion through reaction with NO. Previous measurements of HO$_2$ radicals by instruments using this method
were likely influenced by the conversion of RO$_2$ radicals, with measurements of HO$_2$ in forested environments
likely influenced by interferences from peroxy radicals derived from biogenic alkenes such as isoprene due to the
high RO$_2$-to-HO$_2$ conversion efficiencies of these radicals. Because of the lower conversion efficiencies of alkane-
based peroxy radicals, the impact on previous measurements in urban areas will depend on the relative
concentrations of alkanes versus alkenes and aromatics contributing to the overall pool of peroxy radicals in these
environments.

25       While this interference was taken into account to investigate the radical chemistry during CABINEX

(Griffith et al., 2013) and CalNex (Griffith et al., 2016), this issue was not known when the radical measurements
from the MCMA-2006 field campaign were published (Dusanter et al., 2009b). An analysis of the impact of this
interference on the results for the MCMA-2006 campaign suggests that the RO$_2$ radical contribution to the
measured HO$_2$* concentration was approximately 35% based on the RACM modeled RO$_2$ concentrations. Taking





this interference into account, the resulting modeled $HO_2^*$ concentrations were generally greater than the
measured concentrations by 35% during the afternoon, although the model results were within the calibration
uncertainty of the measurements (36% at $2\sigma$). Given that the modeled $HO_2^*$ concentrations likely reflect a lower
limit to the interference during the campaign these results suggest that the model likely overestimates the measured
concentrations during the afternoon. However, the model still underestimates the $HO_2^*$ concentration by a factor
of 3 in the morning, suggesting that the model may be missing an important radical source in the morning.
Although the measured $HO_2^*/OH$ ratio was in better agreement with the modeled $HO_2^*/OH$ ratio compared to the
modeled $HO_2/OH$ ratio, the model still significantly underestimates the $HO_2^*/OH$ ratio by up to a factor of 4 for
NO mixing ratios greater than 10 ppb, suggesting that our understanding of radical propagation under these
conditions is still incomplete.
Future measurements of peroxy radicals by the IU-FAGE instrument will involve measurements at lower
NO concentrations to minimize the $RO_2$-to-$HO_2$ conversion efficiency. Recent experiments have demonstrated
that at an $HO_2$-to-OH conversion efficiency of approximately 17%, the conversion efficiency of isoprene-based
peroxy radicals is less than 10%. Even at this low $HO_2$-to-OH conversion efficiency, the resulting $HO_2$ signals are
still significantly greater than the limit of detection of the instrument, allowing for measurements of ambient $HO_2$
concentrations without interferences from $RO_2$ radicals.

Acknowledgements. This work was supported by grants from the National Science Foundation (AGS-1104880 and AGS-1440834) and the National Aeronautics and Space Administration (NNX12AE55G). We would also like to thank Stephen Griffith for his helpful insights during the early stages of this project and Jennifer Liljegren for experimental assistance.



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



**Table 1.** Configuration of the IU-FAGE instrument during various previous field campaigns.

|  | CalNex | CABINEX | MCMA-2006 |
| --- | --- | --- | --- |
| Cell pressure (Torr) | 4.1 | 7.5 | 5.4 |
| Orifice diameter (mm/in) | 0.64/0.025 | 1.02/0.04 | 1.02/0.04 |
| Sample flow rate (SLPM) | 3.4 | 8.5 | 10 |
| NO (molecules/cm$^3$) | $2.9 \times 10^{13}$ | $2.1 \times 10^{13}$ | $1.3 \times 10^{13}$ |



**Table 2.** Average measured $f_{RO2\rightarrow HO2}$ for various alkenes and alkanes under different inlet conditions. Uncertainties represents the standard error of the mean from all individual experiments, with the number of experiments shown in parentheses.

| Compounds | 4 Torr @ 3.4 SLPM (CalNex) | 7.5 Torr @ 8.5 SLPM (CABINEX) | 5 Torr @ 10 SLPM (MCMA 2006) | Fuchs et al.[a] | Whalley et al.[b] |
|---|---|---|---|---|---|
| C(HO$_2$→OH) | 0.67 ± 0.01 (67) | 0.90 ± 0.02 (47) | 0.80 ± 0.01 (81) | — | — |
| Isoprene | 0.83 ± 0.07 (5) | 0.91 ± 0.05 (9) | 0.84 ± 0.05 (6) | 0.79 ± 0.05 | 0.92 ± 0.04 |
| MVK | 0.91 ± 0.04 (10) | 0.62 ± 0.05 (21) | 0.72 ± 0.04 (15) | 0.60 ± 0.06 | — |
| MACR | 0.54 ± 0.04 (4) | 0.30 ± 0.07 (5) | 0.32 ± 0.07 (11) | 0.58 ± 0.17 | — |
| MEK | 0.57 ± 0.06 (6) | 0.62 ± 0.01 (2) | 0.51 ± 0.07 (9) | — | — |
| Ethene | 0.65 ± 0.05 (18) | 0.81 ± 0.06 (7) | 0.86 ± 0.06 (9) | 0.85 ± 0.05 | 1.00 ± 0.08 |
| *trans*-2-butene | 0.92 ± 0.04 (4) | — | 0.72 ± 0.03 (6) | — | — |
| TME | 0.96 ± 0.06 (2) | — | 0.64 ± 0.06 (8) | — | — |
| Toluene | 0.65 ± 0.07 (4) | — | 0.32 ± 0.10 (6) | — | — |
| Propane | 0.15 ± 0.03 (5) | — | 0.22 ± 0.11 (2) | — | 0.03 ± 0.01 |
| n-butane | 0.31 ± 0.03 (4) | 0.30 ± 0.03 (3) | 0.23 ± 0.05 (4) | — | 0.18 ± 0.01 |
| n-octane | 0.62 ± 0.04 (5) | — | 0.30 ± 0.05 (5) | — | — |

[a]Fraction of conversion for RO$_2$ to HO$_2$ conversion for the Julich LIF instrument.(Fuchs et al., 2011)
[b]Conversion efficiencies of RO$_2$ to OH for the Leeds LIF instrument referenced to ethene (Whalley et al., 2013)

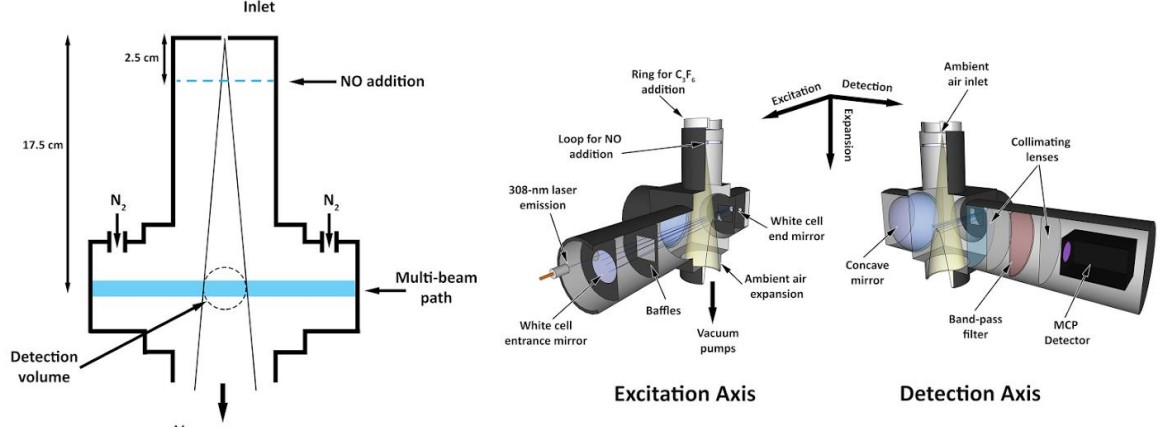

**Figure 1.** Indiana University LIF-FAGE cross section (left) and a schematic of the sampling/excitation axis and the sampling detection axis (right) (Dusanter et al., 2008)

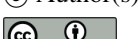


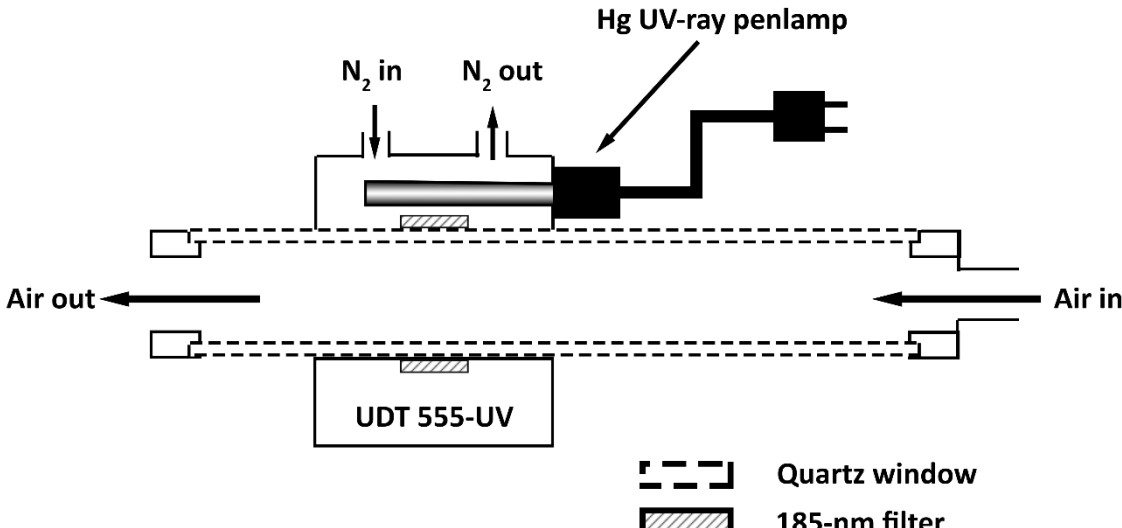

**Figure 2.** Cross-section of Indiana University calibration source for the IU-FAGE instrument





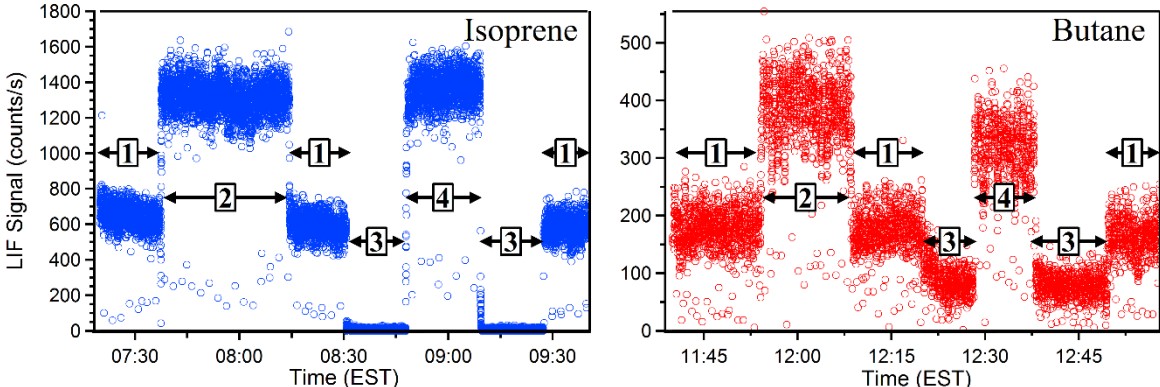

**Figure 3.** RO$_2$ interference measurement experiment for isoprene (left—with an OH reactivity of approximately 290 s$^{-1}$) and butane (right—with an OH reactivity of approximately 30 s$^{-1}$). The boxed numbers within the figure represents the various experimental modes: (1) S$_{OH}$, (2) S$_{HOx}$ with internal NO addition, (3) S$_{OH + VOC}$ with VOC added, (4) S$_{ROx}$ with VOC added and internal NO addition.





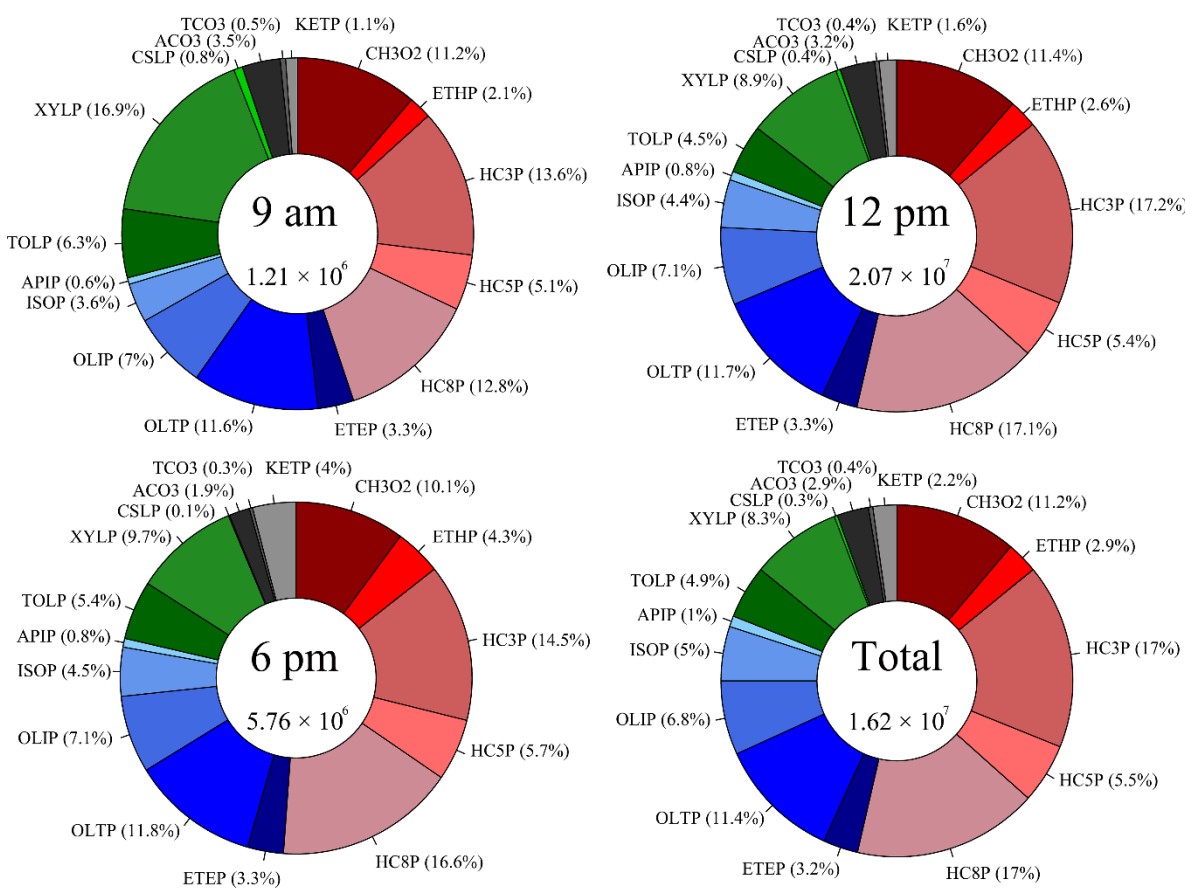

**Figure 4.** Modeled average peroxy radical contributions for the MCMA 2006 field campaign at 9:00 am (top left), 12:00 pm (top right), 6:00 pm (bottom left), and for the average campaign (bottom right). Shades of red represent alkanes, shades of blue represent alkenes, shades of green represent aromatics, and shades of grey represent acyl peroxy radicals.

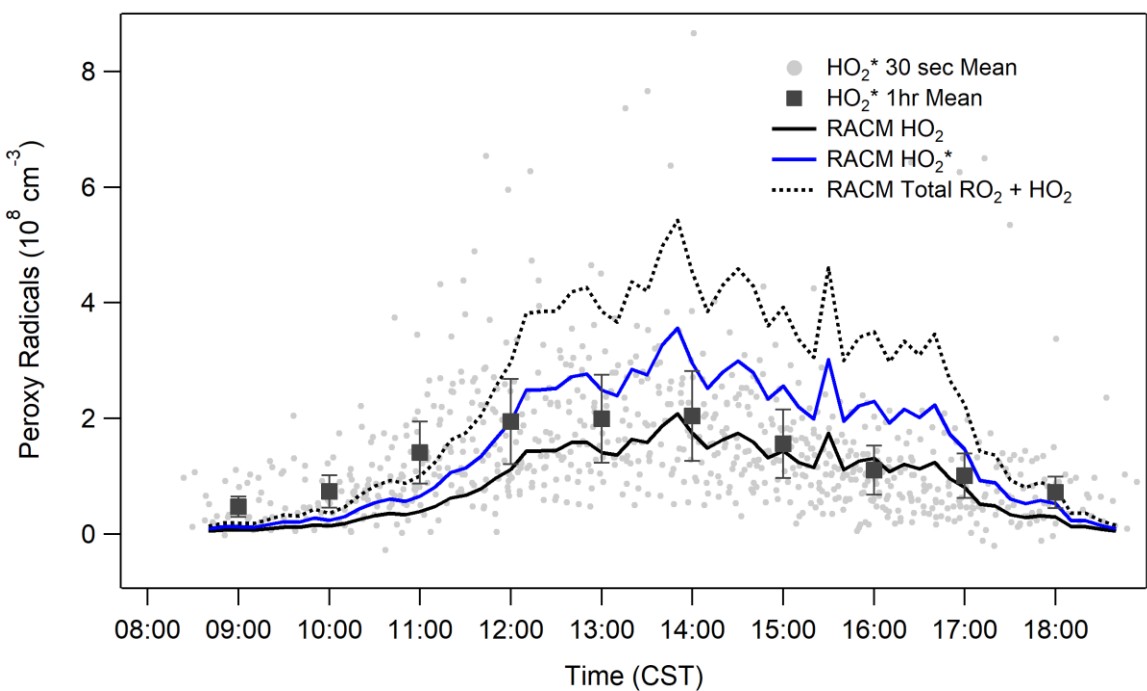

**Figure 5.** Diurnal average HO₂* measurements from MCMA 2006. The grey solid circles are 30 sec averages and solid square symbols are binned 1 hour averages. The solid black line represents the RACM modeled HO₂, the solid blue line represents the modeled HO₂*, and the dotted black line represents the total modeled RO₂ + HO₂. The error bars reflect the calibration accuracy of the measurements (± 36 %, 2σ).



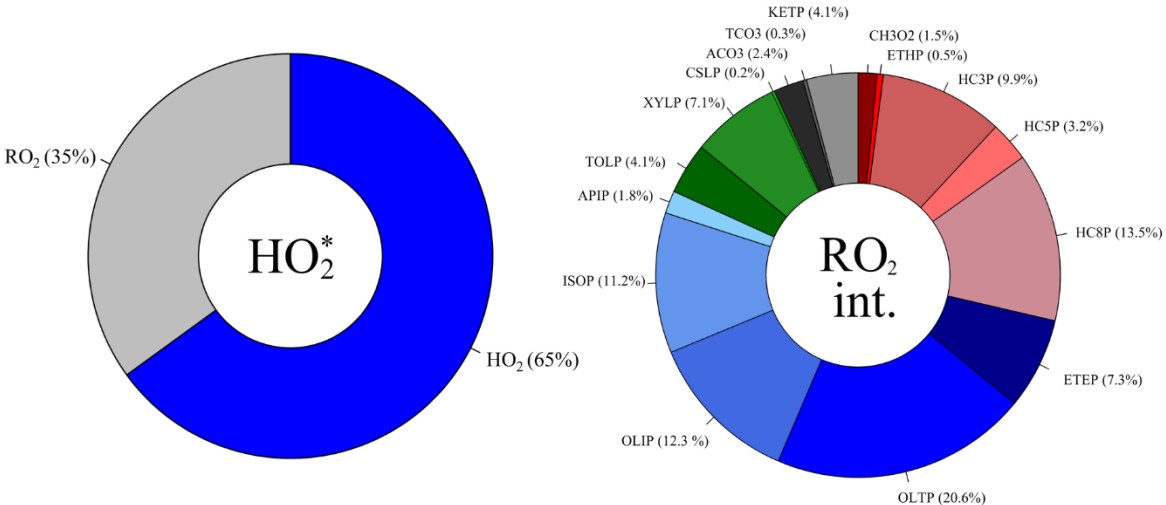

**Figure 6.** Modeled speciation of the RO₂ interference for MCMA 2006. The pie chart on the left is the modeled HO₂*
composition after adding the fraction of RO₂ interference to the modeled HO₂. The pie chart on the right is the composition of
the RO₂ interference.





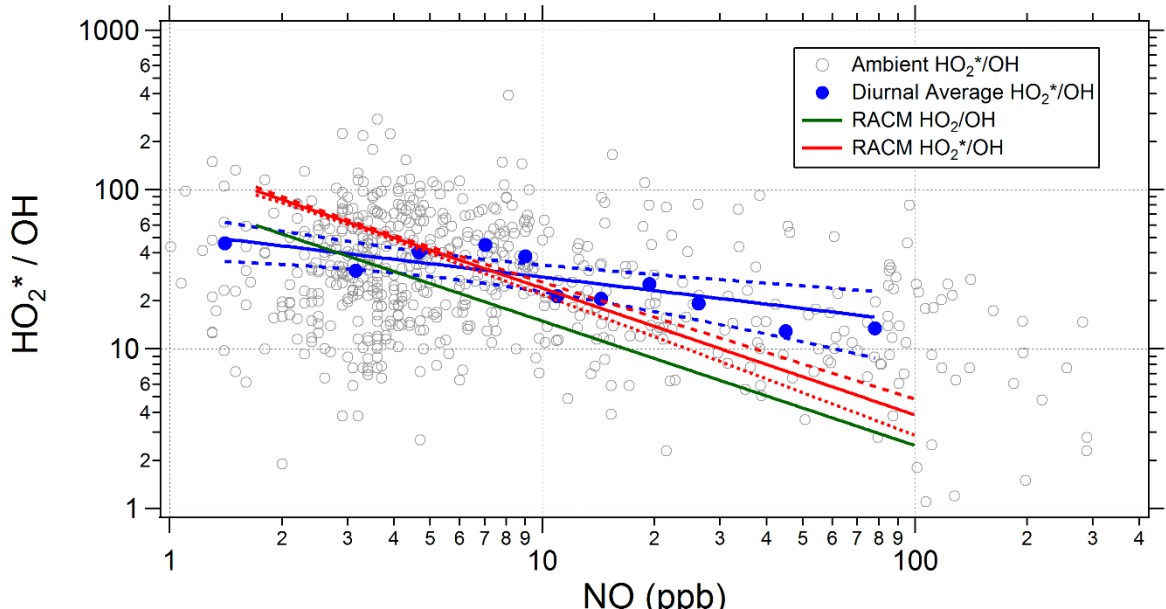

**Figure 7.** Correlation plot for $HO_2*/OH$ vs. NO. Small grey circles are individual measurements recorded for the whole campaign. Large blue circles are average values calculated on binned NO data and the blue line is a fit to the average measurements. The model-calculated $HO_2/OH$ ratio is displayed by the green line for the campaign averaged measurements, while the red line represents the modeled $HO_2*/OH$ ratio. Dashed lines are the 95% confidence interval from the non-linear power regressions.