# Peer review of "Measurement of interferences associated with the detection of the hydroperoxy radical in the atmosphere using laser-induced fluorescence"

_Atmospheric Measurement Techniques, 2017_

## Referee Comment (RC1) · Anonymous Referee #2 · 27 Aug 2017

This manuscript reports laboratory results of interferences from organic peroxy radicals (RO2) on HO2 measurements done by the well accepted FAGE technique. The RO2 interference were studied for another two instruments by Fuchs et al. (2011) and Whalley et al. (2013). Still, the characterization of interference is fundamental for each instrument using chemical conversion, because the relative interference from RO2 towards the HO2 signal will be quite dependent on the individual set up, with NO concentration, reaction time and efficiency of mixing of NO into the flow. The manuscript is well structured and the points are clear. The results are of interest to the community. Therefore,

the referee support the publication in AMT.

Minor comments:

1) The discussion of RO2 interference are mainly associated with the MCMA-2006 campaign. However, the characterization was done with 1 sccm NO addition, which is lower than the flow rates used in the MCMA-2006 campaign. The authors stated that the conversion efficiencies shown in table 2 should be regarded as a lower limit. Could it be possible to quantify how large difference could be made if larger NO flow is used. Using the actual conversion efficiencies will help to discuss the implication of RO2 interferences for HO2 measurements during the MCMA-2006 campaign.

2)The subtraction of HO2 interferences requires the knowledge of speciated RO2 concentrations. Modelled RO2 concentrations could be used as in the present paper, but this would be a dangerous exercise given the likely uncertainties in the model. Could the authors provide the error analysis in the modelled RO2. In fact, RO2 measurements was achieved using LIF technique in a recent field campaign in China, which was higher than model predicted for high NOx conditions but in god agreement in moderate and low NOx regime (Tan et al. 2017 ACP). More discussion should be added if one need to correct the HO2 interferences.

3)One suggestion for further field application and maybe also helpful to the readers. The authors could add a paragraph to describe how to minimize or quantify interference for further field campaigns.

Technical comments:

Page 9, line 15: 'Fig. 2' should be 'Table 2'

Page 9, line 24: after the lower NO concentration adding '(table 1, add the residence time for different cell conditions)'

Page 10, line 15: 'could contribute to the higher RO2-to-HO2 conversion efficiency reported here for MVK' is confusing, suggest to quantify such effect with specific numbers.

Page 10, from line 19 to line 21: the sentence is too long and hard to understand, suggest rephrase it.

Page 10, from line 21 to line 23: It states that the alkoxy radicals isomerize and decompose. Could the author provide reference for it?

---

## Referee Comment (RC2) · Anonymous Referee #1 · 30 Sep 2017

al.

**Anonymous Referee #1**

This paper focuses on the characterization of the RO2-originated interference in the
HO2 signal measured with the LIF-FAGE instrument from the Indiana University. This
interference was shown to affect LIF-FAGE instruments from several groups (Fuchs
et al., 2011;Whalley et al., 2013) in a different amount connected to the geometry of
the detection cell, the methodology of the NO injection and the sample flow. These
together determine the concentration and the mixing of the NO in the cell and affect
the conversion of RO2 into HO2. In this study, several VOCs, relevant for the different
campaigns in which the instrument was deployed, were tested and the impact on the

[Figure]

MCMA-2006 campaign was evaluated.

The paper is well written and the results are well presented. Publication is recommended after the authors address the following points:

1- It is not clear why it was not possible to replicate the exact same NO flow observed during the MCMA-2006 campaign. The authors say that the flow of NO during the test was kept at 1 sccm as this was the flow during all the campaigns (page 8 and 9) although saying that in reality the flow during the MCMA-2006 campaign was changed and a larger flow of NO could (reasonably) explain the discrepancy in the HO2 to OH conversion efficiency observed. Is it not possible to actually operate at the NO flow used during the MCMA-3006 campaign? How different was the NO flow? As this discussion focus on the MCMA-2006 campaign a better characterization of the interference impact for this campaign would be beneficial.

2- The use of RACM to compare with previous results is interesting although, as there is now the availability of RACM2 (which should be an improved version of RACM) and as the authors do mention that the discrepancy between the model results and the measured HO2* could be due to the different treatment of dycarbonyl species, a model run using the more update RACM2 should be performed. It would be an interesting add up to this work and could help understanding the reasons of the discrepancy between model results and measured data.

Minor comments:

Page 4, lines 16 and 20. The laser was changed between the campaigns and the laboratory tests although the name given for the new laser model is the same as for the old laser model. What is the difference then?

Page 5, line 4. Is there any improvement in injecting NO so far from the detection cell? As far as the reviewer is aware most of the other LIF-FAGE instrument inject the NO immediately on the top of the detection cell also to reduce the losses of OH radicals.

Page 7, line 15. It could be helpful to rename COH+VOC in COH→RO2 for consistency with all the other conversion efficiency.

Page 9, line 16. Please, state in the summary table 2 the number of experiments performed for each VOC.

Page 10, lines 10 to 16. It is interesting to observe such a different result from what observed previously by Fuchs et al. (2011). It would be beneficial to extend the discussion a little bit. Why the authors think there is this discrepancy? Is the same type of mercury lamp used by both groups? Could it be possible that the signal observed arises from impurities present in the VOC samples? How much is the HO2 signal due to the photolysis of the VOC?

Page 10, lines 24 to 31. Also here it could be beneficial to extend the discussion. Do the authors have any hypothesis of what could be impacting the conversion of RO2 to HO2 in addition to the points already mentioned?

Page 13, line 15 to 16. The term contrast in this case is misleading. As the authors underling later in the text, the two campaigns are characterized by different VOCs load (one is a forest environment, the other is a city) therefore it is not unexpected to observe a different amount of interference. The sentence should be rephrased. A small paragraph underlying the main chemical conditions for the three campaigns discussed in this work should be add to help the reader understanding similarity and differences between the environments.

Page 22, Table 1. Use Pascal instead of Torr. Remove the inches unit.

Page 27, Figure 4. I suggest grouping the RO2 and use of a more easily understandable labels.

Page 28, Figure 5. The colors of the plot are not easy to separate, I suggest changing the colors.

References:

Fuchs, H., Bohn, B., Hofzumahaus, A., Holland, F., Lu, K. D., Nehr, S., Rohrer, F., and Wahner, A.: Detection of HO2 by laser-induced fluorescence: calibration and interferences from RO2 radicals, Atmos. Meas. Tech., 4, 1209-1225, 10.5194/amt-4-1209-2011, 2011.

Whalley, L. K., Blitz, M. A., Desservettaz, M., Seakins, P. W., and Heard, D. E.: Reporting the sensitivity of laser-induced fluorescence instruments used for HO2 detection to an interference from RO2 radicals and introducing a novel approach that enables HO2 and certain RO2 types to be selectively measured, Atmos. Meas. Tech., 6, 3425-3440, 10.5194/amt-6-3425-2013, 2013.

---

## Author Response (AR1)

**Response to Anonymous Referee #1**

We would like to thank the reviewers for their efforts in reviewing this manuscript, and we feel that the manuscript is much stronger with the suggested changes. Below are detailed responses to their comments, which are highlighted in italics.

*This paper focuses on the characterization of the RO2-originated interference in the HO2 signal measured with the LIF-FAGE instrument from the Indiana University. This interference was shown to affect LIF-FAGE instruments from several groups (Fuchs et al., 2011; Whalley et al., 2013) in a different amount connected to the geometry of the detection cell, the methodology of the NO injection and the sample flow. These together determine the concentration and the mixing of the NO in the cell and affect the conversion of RO2 into HO2. In this study, several VOCs, relevant for the different campaigns in which the instrument was deployed, were tested and the impact on the MCMA-2006 campaign was evaluated.*

*The paper is well written and the results are well presented. Publication is recommended after the authors address the following points:*

*1- It is not clear why it was not possible to replicate the exact same NO flow observed during the MCMA-2006 campaign. The authors say that the flow of NO during the test was kept at 1 sccm as this was the flow during all the campaigns (page 8 and 9) although saying that in reality the flow during the MCMA-2006 campaign was changed and a larger flow of NO could (reasonably) explain the discrepancy in the HO2 to OH conversion efficiency observed. Is it not possible to actually operate at the NO flow used during the MCMA-3006 campaign? How different was the NO flow? As this discussion focus on the MCMA-2006 campaign a better characterization of the interference impact for this campaign would be beneficial.*

The NO flow of 1 sccm used in these experiments is the same NO flow reportedly used during the MCMA 2006 campaign. However, the measured $HO_2$-to-OH conversion efficiency at this NO flow in these experiments was found to be approximately 20% lower than the $HO_2$-to-OH conversion efficiency measured during the MCMA 2006 campaign, and the reason for this discrepancy is unclear. As discussed in the manuscript, potential explanations include the possibility that the NO flow during MCMA-2006 was actually greater than the 1 sccm that was measured, or it may indicate problems in accurately recreating the overall detection cell flow and mixing conditions during this campaign in these laboratory experiments. Since it is not known whether the flow was greater than the 1 sccm that was measured, or whether the flow conditions led to more efficient mixing, we chose to conduct the experiments using the measured 1 sccm flow rate, with the caveat that the conversion efficiencies may represent a lower limit to the actual conversion efficiencies during MCMA-2006. We have clarified
this in the revised manuscript.

*2- The use of RACM to compare with previous results is interesting although, as there is now the*
*availability of RACM2 (which should be an improved version of RACM) and as the authors do*
*mention that the discrepancy between the model results and the measured HO2\* could be due to the*
*different treatment of dycarbonyl species, a model run using the more update RACM2 should be*
*performed. It would be an interesting add up to this work and could help understanding the reasons of*
*the discrepancy between model results and measured data.*

We chose to compare the measurements to the model results using the RACM mechanism so that they
could be compared to the RACM results originally published in Dusanter et al. (2009b). We agree that
a comparison of the measurements with the updated RACM2 mechanism would be valuable, and we
are planning to do this in a subsequent publication that also examines photochemical production rates
of ozone during the campaign.

*Minor comments:*

*Page 4, lines 16 and 20. The laser was changed between the campaigns and the laboratory tests*
*although the name given for the new laser model is the same as for the old laser model. What is the*
*difference then?*

Although the pump lasers are identical, the original laser system operated at 5 kHz repetition rate and
pumped a Lambda Physik dye laser.  Preliminary measurements of the conversion efficiencies were
done with this laser, which was the same model used during the MCMA-2006 campaign. The new
laser operates at 10 kHz and pumps a different dye laser (Sirah Credo).  The conversion efficiencies
measured by the two laser systems were similar.  This has been clarified in the revised manuscript.

*Page 5, line 4. Is there any improvement in injecting NO so far from the detection cell? As far as the*
*reviewer is aware most of the other LIF-FAGE instrument inject the NO immediately on the top of the*
*detection cell also to reduce the losses of OH radicals.*

The longer inlet was originally used to increase the reaction time for the conversion of $HO_2$ to OH.
This inlet length does have the advantage of raising the inlet farther above the detection cell to avoid
any possible surface effects from the environmental cover over the detection cell.  However, it does
lead to increased wall loss of OH which results in somewhat lower sensitivity to ambient OH.

*Page 7, line 15. It could be helpful to rename COH+VOC in COH→RO2 for consistency with all the*
*other conversion efficiency.*

This has been renamed as suggested.

*Page 9, line 16. Please, state in the summary table 2 the number of experiments performed for each VOC.*

The number of experiments performed for each VOC is given by the number in parenthesis after the conversion efficiency in Table 2.

*Page 10, lines 10 to 16. It is interesting to observe such a different result from what observed previously by Fuchs et al. (2011). It would be beneficial to extend the discussion a little bit. Why the authors think there is this discrepancy? Is the same type of mercury lamp used by both groups? Could it be possible that the signal observed arises from impurities present in the VOC samples? How much is the HO2 signal due to the photolysis of the VOC?*

We have expanded the discussion as suggested, as differences in the mercury lamp flux or impurities in the VOC samples could have led to the production of both OH and $HO_2$ radicals from the photolysis of these VOCs. We have added tables to the Supplementary Information that describes the purity of the VOCs used in these experiments.

*Page 10, lines 24 to 31. Also here it could be beneficial to extend the discussion. Do the authors have any hypothesis of what could be impacting the conversion of RO2 to HO2 in addition to the points already mentioned?*

As suggested, we have expanded the discussion of the mechanism of peroxy radical decomposition to $HO_2$ from the OH-initiated oxidation of MVK and MACR. As pointed out by Fuchs et al. (2011) the fates of the peroxy radicals produced by the OH initiated oxidation of MVK and MACR are not well known and likely involve multiple channels with different reaction times, resulting in a more complex dependence on reaction time compared to the mechanism of $HO_2$ production from other alkenes.

*Page 13, line 15 to 16. The term contrast in this case is misleading. As the authors underling later in the text, the two campaigns are characterized by different VOCs load (one is a forest environment, the other is a city) therefore it is not unexpected to observe a different amount of interference. The sentence should be rephrased. A small paragraph underlying the main chemical conditions for the three campaigns discussed in this work should be add to help the reader understanding similarity and differences between the environments.*

We have rephrased this sentence as suggested, focusing on how the different environments lead to different contributions of the various peroxy radicals to the overall interference during each campaign. As part of this rephrasing, we have added information on the relative contribution of individual peroxy radicals to the overall interference for each campaign, providing additional information on the similarity and differences between these environments.

*Page 22, Table 1. Use Pascal instead of Torr. Remove the inches unit.*

This has been changed as suggested.

*Page 27, Figure 4. I suggest grouping the RO2 and use of a more easily understandable labels.*

We chose to include the RACM category labels for each peroxy radical in this plot to illustrate the
contribution of each RACM peroxy radical category to the total modeled peroxy radical
concentration.  The RACM labels are defined in Section 4.2 (page 13 of the revised manuscript), and
we have revised the caption to help clarify these points.

*Page 28, Figure 5. The colors of the plot are not easy to separate, I suggest changing the colors.*

We have changed the colors of the plot as suggested.

**Response to Anonymous Referee #2**

We would like to thank the reviewers for their efforts in reviewing this manuscript, and we feel that the manuscript is much stronger with the suggested changes. Below are detailed responses to their comments, which are highlighted in italics.

*This manuscript reports laboratory results of interferences from organic peroxy radicals (RO2) on HO2 measurements done by the well accepted FAGE technique. The RO2 interference were studied for another two instruments by Fuchs et al. (2011) and Whalley et al. (2013). Still, the characterization of interference is fundamental for each instrument using chemical conversion, because the relative interference from RO2 towards the HO2 signal will be quite dependent on the individual set up, with NO concentration, reaction time and efficiency of mixing of NO into the flow. The manuscript is well structured and the points are clear. The results are of interest to the community. Therefore, the referee support the publication in AMT.*

*Minor comments:*

*1) The discussion of RO2 interference are mainly associated with the MCMA-2006 campaign. However, the characterization was done with 1 sccm NO addition, which is lower than the flow rates used in the MCMA-2006 campaign. The authors stated that the conversion efficiencies shown in table 2 should be regarded as a lower limit. Could it be possible to quantify how large difference could be made if larger NO flow is used. Using the actual conversion efficiencies will help to discuss the implication of RO2 interferences for HO2 measurements during the MCMA-2006 campaign.*

We have clarified that the NO flow of 1 sccm used in these experiments is the same NO flow reportedly used during the MCMA 2006 campaign. However, the measured $HO_2$-to-OH conversion efficiency at this NO flow in these experiments was found to be approximately 20% lower than the $HO_2$-to-OH conversion efficiency measured during the MCMA 2006 campaign, and the reason for this discrepancy is unclear. As discussed in the manuscript, potential explanations include the possibility that the NO flow during MCMA-2006 was actually greater than the 1 sccm that was measured, or it may indicate problems in accurately recreating the flow conditions during this campaign in these laboratory experiments. Since it is not known whether the flow was greater than the 1 sccm that was measured, or whether the flow conditions during the campaign led to more efficient mixing, we chose to conduct the experiments using the measured 1 sccm flow rate, with the caveat that the conversion efficiencies may represent a lower limit to the actual conversion efficiencies during MCMA-2006. Given that the conversion efficiencies for the other instrumental configurations do not appear to directly correlate with the measured $HO_2$-to-OH conversion efficiency, it is difficult to quantify how the higher $HO_2$-to-OH conversion efficiency measured during MCMA-2006 would translate into the various $RO_2$-to-$HO_2$ conversion efficiencies, although it is likely that many of them
would be larger. We have attempted to clarify this in section 3 of the revised manuscript.

*2) The subtraction of HO2 interferences requires the knowledge of speciated RO2 concentrations.*
*Modelled RO2 concentrations could be used as in the present paper, but this would be a dangerous*
*exercise given the likely uncertainties in the model. Could the authors provide the error analysis in*
*the modelled RO2. In fact, RO2 measurements was achieved using LIF technique in a recent field*
*campaign in China, which was higher than model predicted for high NOx conditions but in god*
*agreement in moderate and low NOx regime (Tan et al. 2017 ACP). More discussion should be added*
*if one need to correct the HO2 interferences.*

As pointed out by the reviewer, it is possible to correct the measured $HO_2$* through subtraction of the
modeled speciated $RO_2$ interferences, and compare these results to the modeled $HO_2$ concentrations.
However, as noted by the reviewer, this method has a much greater uncertainty as a result of the
uncertainty associated with the modeled $RO_2$ measurements. We estimate that the uncertainty
associated with the modeled $RO_2$ to be approximately $\pm70\%$ ($2\sigma$), similar to that for the modeled $HO_2$
(Dusanter et al., 2009b). As a result, we prefer to compare the modeled $HO_2$* to the measured $HO_2$*.
This has been clarified in the revised manuscript.

As suggested, we have also included a discussion of the results from Tan et al. (2017) regarding the
model underestimation of their morning $RO_2$ measurements, which appear to be consistent with the
morning observations during MCMA-2006.

*3) One suggestion for further field application and maybe also helpful to the readers. The authors*
*could add a paragraph to describe how to minimize or quantify interference for further field*
*campaigns.*

We have expanded the discussion of minimizing the $RO_2$ interference in section 5 as suggested,
including more quantitative information on the concentration of NO that we have shown to minimize
the interference from isoprene-based peroxy radicals.

*Technical comments:*

*Page 9, line 15: 'Fig. 2' should be 'Table 2'*

This typo has been corrected, as the text is actually referring to the experiments shown in Fig. 3.

*Page 9, line 24: after the lower NO concentration adding '(table 1, add the residence time for*
*different cell conditions)'*

We have added the reference to Table 1 as suggested. The reaction time for the different
configurations is approximately 1-2 ms based on simulations of the kinetics of the system.

Unfortunately, the precise residence time for the different flow conditions is difficult to simulate
given the different OH radical wall losses that may be occurring.

*Page 10, line 15: 'could contribute to the higher RO2-to-HO2 conversion efficiency reported here for*
*MVK' is confusing, suggest to quantify such effect with specific numbers.*

We have expanded and clarified the discussion of this potential interference as suggested. However,
the actual interference is difficult to quantify as addition of water vapor may reduce the $HO_x$ radical
production from photolysis of these VOCs through quenching of the excited VOC.

*Page 10, from line 19 to line 21: the sentence is too long and hard to understand, suggest rephrase it.*

We have shortened and rephrased this sentence as suggested.

*Page 10, from line 21 to line 23: It states that the alkoxy radicals isomerize and decompose. Could*
*the author provide reference for it?*

We have provided references as suggested (Atkinson, R., Int. J. Chem. Kinet., 29, 99-111, 1997;

[revised manuscript text omitted]
$) | $0.67 \pm 0.01$ (67) | $0.90 \pm 0.02$ (47) | $0.80 \pm 0.01$ (81) | — | — |
| Isoprene | $0.83 \pm 0.07$ (5) | $0.91 \pm 0.05$ (9) | $0.84 \pm 0.05$ (6) | $0.79 \pm 0.05$ | $0.92 \pm 0.04$ |
| MVK | $0.91 \pm 0.04$ (10) | $0.62 \pm 0.05$ (21) | $0.72 \pm 0.04$ (15) | $0.60 \pm 0.06$ | — |
| MACR | $0.54 \pm 0.04$ (4) | $0.30 \pm 0.07$ (5) | $0.32 \pm 0.07$ (11) | $0.58 \pm 0.17$ | — |
| MEK | $0.57 \pm 0.06$ (6) | $0.62 \pm 0.01$ (2) | $0.51 \pm 0.07$ (9) | — | — |
| Ethene | $0.65 \pm 0.05$ (18) | $0.81 \pm 0.06$ (7) | $0.86 \pm 0.06$ (9) | $0.85 \pm 0.05$ | $1.00 \pm 0.08$ |
| *trans*-2-butene | $0.92 \pm 0.04$ (4) | — | $0.72 \pm 0.03$ (6) | — | — |
| TME | $0.96 \pm 0.06$ (2) | — | $0.64 \pm 0.06$ (8) | — | — |
| Toluene | $0.65 \pm 0.07$ (4) | — | $0.32 \pm 0.10$ (6) | — | — |
| Propane | $0.15 \pm 0.03$ (4) | — | $0.22 \pm 0.11$ (2) | — | $0.03 \pm 0.01$ |
| n-butane | $0.31 \pm 0.03$ (4) | $0.30 \pm 0.03$ (3) | $0.23 \pm 0.05$ (4) | — | $0.18 \pm 0.01$ |
| n-octane | $0.62 \pm 0.04$ (5) | — | $0.30 \pm 0.05$ (5) | — | — |

[a] Fraction of conversion for $RO_2$ to $HO_2$ conversion for the Julich LIF instrument (Fuchs et al., 2011)
[b] Conversion efficiencies of $RO_2$ to OH for the Leeds LIF instrument referenced to ethene (Whalley et al., 2013)

[Figure]

**Figure 1.** Indiana University LIF-FAGE cross section (left) and a schematic of the sampling/excitation axis and the sampling detection axis (right) (Dusanter et al., 2008)

[Figure]

**Figure 2.** Cross-section of Indiana University calibration source for the IU-FAGE instrument

[Figure]

**Figure 3.** $RO_2$ interference measurement experiment for isoprene (left—with an OH reactivity of approximately 290 s$^{-1}$) and butane (right—with an OH reactivity of approximately 30 s$^{-1}$). The boxed numbers within the figure represents the various experimental modes: (1) $S_{OH}$, (2) $S_{HOx}$ with internal NO addition, (3) $S_{OH + VOC}$ with VOC added, (4) $S_{ROx}$ with VOC added and internal NO addition.

[Figure]

**Figure 4.** Modeled average peroxy radical contributions for the MCMA 2006 field campaign at 9:00 am (top left), 12:00 pm (top right), 6:00 pm (bottom left), and for the average campaign (bottom right). Shades of red represent alkanes, shades of blue represent alkenes, shades of green represent aromatics, and shades of grey represent acyl peroxy radicals. Individual RACM peroxy radical categories are defined in Section 4.2.

[Figure]

**Figure 5.** Diurnal average $HO_2^*$ measurements from MCMA 2006. The grey solid circles are 30 sec averages and solid blue square symbols are binned 1 hour averages. The solid black line represents the RACM modeled $HO_2$, the solid red line represents the modeled $HO_2^*$, and the dotted black line represents the total modeled $RO_2 + HO_2$. The error bars reflect the calibration accuracy of the measurements ($\pm 36$ %, $2\sigma$).

[Figure]

**Figure 6.** Modeled speciation of the RO$_2$ interference for MCMA 2006. The pie chart on the left is the modeled HO$_2$*
composition after adding the fraction of RO$_2$ interference to the modeled HO$_2$. The pie chart on the right is the composition of
the RO$_2$ interference.

[Figure]

**Figure 7.** Correlation plot for $HO_2$*/OH vs. NO. Small grey circles are individual measurements recorded for the whole campaign. Large blue circles are average values calculated on binned NO data and the blue line is a fit to the average measurements. The model-calculated $HO_2$/OH ratio is displayed by the green line for the campaign averaged measurements, while the red line represents the modeled $HO_2$*/OH ratio. Dashed lines are the 95% confidence interval from the non-linear power regressions.

**Measurement of interferences associated with the detection of the hydroperoxy radical in the atmosphere using laser-induced fluorescence**

Michelle M. Lew[1], Sebastien Dusanter[2,3], and Philip S. Stevens[1,3]

(1) Department of Chemistry, Indiana University, Bloomington, IN, USA
(2) IMT Lille Douai, Univ. Lille, SAGE - Département Sciences de l'Atmosphère et Génie de l'Environnement, 59000 Lille, France
(3) School of Public and Environmental Affairs, Indiana University, Bloomington, IN, USA

*Correspondence to*: Philip S. Stevens (pstevens@indiana.edu)

**Supplementary Information**

**Table S1:** Gas-phase compounds used in the $RO_2$ to $HO_2$ interference experiments

| Compound | Company | Conc. | Conc. Cert. | Balance Gas | Blend Tolerance | Certified Accuracy |
|---|---|---|---|---|---|---|
| Isoprene | Matheson | 100 ppm | 99 ppm | $N_2$ | 10 % | 2 % |
| Tetramethyl ethylene | Matheson | 15 ppm | 17.4 ppm | $N_2$ | 20 % | 5 % |
| trans-2-butene | Matheson | 30 ppm | 30.2 ppm | $N_2$ | 20 % | 5 % |
| Ethylene | Matheson | 150 ppm | -- | $N_2$ | -- | -- |
| Ethane | Matheson | 1200 ppm | 1201 ppm | $N_2$ | 10 % | 2 % |
| Propane | Matheson | 1200 ppm | -- | $N_2$ | -- | -- |
| Butane | Matheson | 650 ppm | 650 ppm | $N_2$ | 10 % | 2 % |

**Table S2:** Liquid-phase compounds used in the $RO_2$ to $HO_2$ interference experiments.

| Compound | Company | Purity |
|---|---|---|
| 3-buten-2-one (MVK) | Sigma-Aldrich | 99% |
| Methacrolein (MACR) | Aldrich | 95% |
| 2-butanone (MEK) | Sigma-Aldrich | 99% |
| n-octane | TCI-EP | 97% |
| Toluene | Macron Chemicals | 99.5% |

---

## Author Response (AR2)

Dear Dr. Richter,

Thank you for assuming the role of editor for our manuscript and for your comments, which are highlighted in blue below with our responses and changes.

We hope that with this revisions that the manuscript will be suitable for publication in Atmospheric Measurement Techniques. Please feel free to contact me if you have any additional questions.

Very truly yours,

Philip S. Stevens
James H. Rudy Professor
Indiana University

*Even after your revisions, I'm confused by the discussion of NO flow rates. While you explain in the replies to the reviewers, that the nominal flow rate during MCMA was the same as used in your tests, the last paragraph on page 20 as well as the last sentence on page 21 appear to state something different. Can you please clarify?*

Thank you for pointing out the inconsistencies in this section of the manuscript. We have revised the paragraph on page 20 (page 13 in the revised manuscript) as follows for clarification:

"Because the $HO_2$-to-OH conversion efficiency of 80% in these experiments was approximately 20% lower than the conversion efficiency measured during the campaign (Dusanter et al., 2008; Dusanter et al., 2009a), the relative peroxy radical contributions illustrated in this figure are likely lower limits to the actual contribution during the campaign."

We have also revised the last sentence on page 21 (page 14 of the revised manuscript) as follows:

"As discussed above, the modeled $HO_2*$ is likely a lower limit given that the $RO_2$-to-$HO_2$ conversion efficiencies during the campaign may be greater than shown in Table 2 due to the higher $HO_2$-to-OH conversion efficiency measured during the campaign."

*Why is NO concentration given in table 1? In all other places you discuss NO flow so I'm wondering if you did this intentionally to provide additional information. I assume that these values are just computed from the measured NO flow?*

We included the concentration of NO in this table to provide additional information that could be compared to the NO concentrations reported previously by Fuchs et al. (2011) and Whalley et al. (2012). We have clarified in the table that these concentrations are computed from the measured 1sccm NO flow.

*I'm not convinced that there is a difference in error propagation between adding modelled interferences to modelled HO2 or subtracting modelled interferences from measured HO2* as stated on page 21. I would expect that there is actually no difference either way. Please comment.*

Subtracting the modeled $RO_2$ interference from the measured $HO_2*$ to obtain a measured $HO_2$ concentration would increase the uncertainty associated with the resulting measured $HO_2$ concentration compared to the $HO_2^*$ measurement uncertainty. We have clarified this on page 14 in the revised manuscript as follows:

"While it is possible to subtract the modeled speciated $RO_2$ concentrations from the measured $HO_2^*$ and compare the results to the modeled $HO_2$, this method increases the uncertainty associated with the measured concentrations due to the additional uncertainty associated with the modeled $RO_2$ concentrations as well as the uncertainties associated with the measured $RO_2$–to-$HO_2$ conversion efficiencies."

*page 16, line 12: Thus is possible => Thus it is possible*

This has been corrected on page 9 of the revised manuscript.

*page 18, line 19: duplicate "possible"*

This has been corrected on page 11 of the revised manuscript.

*page 21, line 22: We estimate that ... to be => We estimate ... to be*

This has been corrected on page 14 line 20 of the revised manuscript.

[revised manuscript text omitted]